# Assessment of Three-Dimensional Kinematics of High- and Low-Calibre Hockey Skaters on Synthetic ice Using Wearable Sensors

**DOI:** 10.3390/s23010334

**Published:** 2022-12-28

**Authors:** Aminreza Khandan, Ramin Fathian, Jason P. Carey, Hossein Rouhani

**Affiliations:** Department of Mechanical Engineering, University of Alberta, Edmonton, AB T6G 1H9, Canada

**Keywords:** inertial measurement unit, objective performance assessment, synthetic ice skating, three-dimensional kinematics, supervised classification analysis

## Abstract

Hockey skating objective assessment can help coaches detect players’ performance drop early and avoid fatigue-induced injuries. This study aimed to calculate and experimentally validate the 3D angles of lower limb joints of hockey skaters obtained by inertial measurement units and explore the effectiveness of the on-ice distinctive features measured using these wearable sensors in differentiating low- and high-calibre skaters. Twelve able-bodied individuals, six high-calibre and six low-calibre skaters, were recruited to skate forward on a synthetic ice surface. Five IMUs were placed on their dominant leg and pelvis. The 3D lower-limb joint angles were obtained by IMUs and experimentally validated against those obtained by a motion capture system with a maximum root mean square error of 5 deg. Additionally, among twelve joint angle-based distinctive features identified in other on-ice studies, only three were significantly different (*p*-value < 0.05) between high- and low-calibre skaters in this synthetic ice experiment. This study thus indicated that skating on synthetic ice alters the skating patterns such that the on-ice distinctive features can no longer differentiate between low- and high-calibre skating joint angles. This wearable technology has the potential to help skating coaches keep track of the players’ progress by assessing the skaters’ performance, wheresoever.

## 1. Introduction

One of the key components of ice hockey players’ skills is skating [1,2]. Like other sports activities, skating has traditionally been assessed by video and motion capture cameras [3,4,5]. These cameras have been used to study two-dimensional (2D) or three-dimensional (3D) kinematics of the lower-limb joints of individuals skating. A setup of digital video cameras was used to obtain 2D or 3D joint angles on ice [6,7,8,9] or on a skating treadmill [5,10]. Moreover, in [6,7,8,9], motion capture systems were used to obtain the kinematics of high- and low-calibre or male and female high-calibre hockey players. Although these motion capture systems are precise and taken as a reference system, their application is bounded to in-lab measurements due to their limited availability and capturing volume. Instead, wearable technology is a trended and acclaimed alternative for performance assessment and can be used in in-field measurements [11].

Wearable technologies such as GPS and accelerometers measure essential kinematics in team sports [12,13,14,15,16,17,18]. A 3D accelerometer enables researchers to determine temporal events during ice hockey skating and also differentiate players in terms of their skill levels [16,17,19]. However, neither 3D accelerometers alone nor GPS can measure 3D joint angles. GPS works precisely in open-field sports; however, the signals of GPS may be considerably affected by errors in indoor areas. Moreover, GPS does not provide physiologically relevant information, such as the players’ phase of play during ice hockey. Therefore, despite the wealth of accelerometers and GPS measurements, they cannot provide inclusive biomechanical information for comprehensive on-ice assessments. Instead, inertial measurement unit (IMU) technology can be used for on-ice athletics performance assessment, similar to their established acceptance in clinical outcome evaluations [20,21,22]. Moreover, IMUs’ readout can precisely calculate the joint angles and temporal and spatial parameters of athletic activities [11,23] and potentially differentiate players according to their skill levels.

Biomechanical variances between high- and low-calibre players’ skating have been emphasized to understand the relationship between skating biomechanics and players’ performance. Previous research has highlighted substantial differences between groups of players with different skill levels in ice experiments [6,9,19,24]. They have found significant differences between 3D angles of lower limb joints and body centre of mass (CoM) movements between low- and high-calibre players. In another study, Robbins et al. [25] used principal component analysis (PCA) to extract the most significant features to differentiate high- and low-calibre players’ 3D joint angles in on-ice experiments. In these studies, parameters such as higher ankle plantar flexion, knee extension at push-off, and higher hip flexion were found to be different in high-calibre players’ skating compared to low-calibre players’ skating. These on-ice calibre-based distinctive kinematic features (or simply distinctive features), however, may not be distinctive on synthetic ice.

While most public ice rinks were closed due to COVID-19, synthetic ice showed to be an alternative for skaters to exercise and be prepared for competitions and assists coaches in monitoring their players remotely. Stidwill et al. [26] showed that the gross movement patterns of skating on synthetic ice surfaces were similar to skating on ice. However, they reported differences in the kinematics and postures of the participants skating on synthetic ice compared to ice. Therefore, skating on synthetic ice can also affect the distinctiveness of the on-ice distinctive features between high- and low-calibre hockey skaters (or simply skaters) on synthetic ice.

The objectives of this study are to: (1) calculate the 3D angles of lower limb joints of participants skating using IMUs, (2) experimentally validate the accuracy of the obtained angles against those measured by a motion capture system, and (3) experimentally investigate if the kinematic features of lower limb joints during skating on synthetic ice measured by this wearable system can differentiate low- and high-calibre skaters. The outcome of this technology will be an optimal set of wearable IMU sensors ready to be used in on-ice and on-synthetic-ice experiments to measure the 3D kinematics of the skaters.

## 2. Materials and Methods

### 2.1. Participants

Twelve able-bodied individuals were recruited to participate in this study. By a K-means clustering algorithm based on the participant’s years of ice skating experience, they are clustered into two groups of six high-calibre skaters (age 24 ± 4 years, height 164 ± 3 cm, body mass 66 ± 7 kg, ice skating experience 18 ± 4 years, four female and two male) and six low-calibre skaters (age 24 ± 4 years, height 174 ± 8 cm, body mass 72 ± 11 kg, ice skating experience 6 ± 6 years, two female and four male). All participants were healthy individuals and could skate on synthetic ice without trouble. They were informed of the experimental procedures and gave informed written consent before the test. This study was approved by the University of Alberta’s research ethics board (Pro00092821).

### 2.2. Experiments

The tests were performed on a motion capture lab walkway covered with 14 × 2 m^2^ synthetic ice panels. Four IMUs composed of accelerometers, gyroscopes, and magnetometers (Xsens Technologies (Enschede, The Netherlands) [27], full-scale ranges: acceleration: ±160 m/s^2^, angular velocity: ±2000 deg/s, and magnetic field: ±1.9 Gauss, plate size: 10 × 7 cm) were placed on the pelvis, thighs, shanks, and hockey skates on the dominant legs of the participants without constraining their movements (Figure 1). As the gold standard for 3D joint angles, a motion capture system with 12 infrared cameras (eight Vero and four Bonita, Vicon, UK [28]) was used to obtain the 3D marker trajectories. These markers were placed on the participant’s body’s anatomical landmarks following an experimental protocol suggested by Cappozzo et al. ([29], Figure 1). The participants were asked to skate until they confirmed that they were comfortable skating on synthetic ice. Then, from a stationary position on one side of the synthetic ice surface, they skated forward and stopped at the other end of the surface. This skating trial was repeated five times, and marker trajectories and IMU readout were captured simultaneously at a sampling frequency of 100 Hz.

### 2.3. 3D Joint Angles Validation

IMU readouts, including the tri-axial gyroscope, accelerometer, and magnetometer measurements, are used to obtain the sensor orientations using sensor fusion algorithms [30,31,32]. These orientations are usually subject to uncertainties and errors due to sensors’ biases and noises [32]. The sensor fusion algorithms address these uncertainties and help obtain a more accurate sensor orientation used for 3D joint angle estimation [31,32]. The 3D joint angles obtained in this study were estimated using the sensor fusion algorithm developed and presented in the Xsens software package (MT manager [27]). In this study, the IMU readouts were first filtered using a 4th-order low-pass Butterworth filter with a cut-off frequency of 15 Hz. Then, the sensor orientations (also known as IMU frames) were obtained in lab global reference frames using the retro-reflective markers fixed on the plates (plate orientations) and sensor-to-plate orientations obtained following the procedure suggested in [33,34,35] (Figure 2). Next, the segment orientations were calculated using the corrected sensor orientations of IMUs and sensor-to-body orientation obtained by sensor and segment frames obtained from the plate and anatomical markers [29], respectively, at the beginning of the capturing session [33,35]. Finally, using the obtained segment orientation, ankle, knee, and hip joint angles in the captured trials were calculated and expressed by the joint coordinate system (JCS) [36]. At the same time, the 3D joint angles were obtained using the markers placed on anatomical landmarks. The flowchart of the 3D joint angles calculation using IMU readouts and validation of the angles against those obtained by camera recordings as the reference system is shown in Figure 2.

Finally, the Root mean square (RMS) between the 3D joint angles calculated from the IMUs’ readout and the angles calculated from the reference system were obtained:The RMS errors between IMU-based and camera recordings-based angles for each trial of all participants were calculated;The average of the RMS error was calculated over all trials of each participant;The computed average value for each participant is presented as boxplots for each 3D joint angle.

In the next step, using these validated angles and the on-ice distinctive parameters, a supervised classification analysis was developed to differentiate low- and high-calibre skaters’ profiles using the on-ice distinctive features.

### 2.4. Calibre-Based Classification Analysis

First, we experimentally investigated if the distinctive features measured by this wearable IMU system would be different between low- and high-calibre skaters on synthetic ice. All the temporal events of skating were obtained based on the camera recordings as the reference system in order to avoid mixing errors due to sensor orientation estimation and temporal event detection. Then, a Friedman’s test was used to verify whether the distinctive features obtained from the literature studies (listed in Table 1) significantly differed between high- and low-calibre skaters skating in this experiment. When a *p*-value was lower than or equal to 0.05, the feature was chosen to classify high- and low-calibre skaters in this study. Then, a K-nearest neighbour (KNN) model was used to classify high- and low-calibre skaters (Figure 3). This supervised classifier was selected because of its reliability, simplicity, and small sample size in this study. The KNN was implemented for k = 1 to k = 15 to classify high- and low-calibre skaters using a cross-validation analysis using the following steps: Each of the five trials of the participant was labelled according to the participant’s calibre;Three participants’ data (25% of the data) were randomly selected and added to the testing set, and the other participants’ data (75% of the data) were added to the training set;A KNN classification was trained for k = 1 to k = 15, and the sensitivity, specificity, precision, and accuracy of these classifiers were calculated for the testing set;Steps (1–3) were repeated 12 times so that each participant was added to the testing set three times;The average of all obtained sensitivity, specificity, precision, and accuracy for these repeated measures was calculated for each k.

Alternatively, while preserving the essential information, a principal components analysis (PCA) was used to optimize the feature space dimensions used in the cross-validation analysis. The best three components obtained from the PCA on on-ice distinctive features were selected as the new feature space of the KNN classification, and steps (1–5) were repeated (Figure 3).

## 3. Results

### 3.1. 3D Joint Angles Validation

Sixty measurement trials from the twelve participants, five trials for each, were obtained (Figure 4); one complete skate stride of the dominant leg of the participants is available in each trial. According to Figure 5, the maximum average RMS errors of the lower limb 3D joint angles during skating obtained by IMUs readout against those obtained by camera recordings across different joints was 5 deg (knee external rotation).

### 3.2. Calibre-Based Classification Analysis

Based on a Friedman’s test, only three out of 12 on-ice distinctive features were different between low- and high-calibre skaters skating, based on 3D lower limb joint angles during skating on synthetic ice (Table 1, *p*-value < 0.05): (i) ankle dorsiflexion range, (ii) hip adduction angle in push-off instant, and (iii) knee flexion angle in a push-off instant, referred to hereafter as selected features. When the KNN classification with k varying from 1 to 15 was used based on these three selected features to differentiate high-calibre skaters from low-calibre skaters, the classifiers’ sensitivity, specificity, precision, and accuracy ranged from 46% to 67%, 61% to 86%, 47% to 67%, and 59% to 75%, respectively (Table 2). Additionally, using PCA’s first three principal components implemented on these three selected features, the classifiers’ sensitivity, specificity, precision, and accuracy ranged from 46% to 71%, 58% to 78%, 53% to 58%, and 68% to 74%, respectively (Table 2).

## 4. Discussion

The 3D joint angles of ice skating were obtained by wearable IMUs and cross-validated against the angles obtained by a motion capture system. The accuracy of the 3D joint angles in synthetic ice skating (less than 5 deg of error) was comparable with those reported in gait analysis (1 to 4 deg [22,37]). The slightly higher errors compared to gait analysis can be due to the higher range of motions of lower limb joints in ice skating.

The 3D joint angles of the lower limbs of low- and high-calibre skater groups showed significant differences in several distinctive features (listed in Table 1). Nevertheless, in our experiments on synthetic ice, only one-fourth of these on-ice distinctive features showed a significant difference between lower limb joint angles of high- and low-calibre skaters. For instance, hip adduction at initial contact—known to be effective in most on-ice studies [19,24,25]—was not significantly different between high- and low-calibre skaters in this study. We concluded that skating on synthetic ice altered the kinematics of the participants’ lower limbs compared to ice such that most of the on-ice distinctive features were no longer different between low- and high-calibre skaters on synthetic ice. Concurrently, it was shown that skating on synthetic ice changes the skaters’ kinematics and temporal and spatial parameters compared to on-ice skating [26,38] and lack of familiarity, shorter skating distance, and different surface friction coefficients were introduced as the possible factors:Lack of familiarity with skating on synthetic ice may have affected the skating patterns of the participants since most of the study’s participants, even high-calibre ones, had not had much experience with synthetic ice skating prior to the skating sessions.Skating on a shorter distance—here, a 14 m-distance—requires faster acceleration and deceleration than real ice skating [26], which could affect the skating patterns of the participants on the synthetic surfaces.The reported surface-skates blades friction coefficient of the synthetic surfaces (0.27–0.37) is higher than the reported ice skating surfaces (0.002–0.007) [26,39,40,41].

Furthermore, even the three best distinctive features could not increase the accuracy and sensitivity of the KNN models by more than 64% and 53%, respectively. Even the first three principal components obtained from the feature space could not increase the accuracy and sensitivity by more than 65% and 53%, respectively, which are almost the same as the previously selected feature space. Therefore, a newly updated feature space extracted from a larger synthetic ice-skating sample—from both male and female ice skaters—is required in future studies to achieve improved KNN performance on these synthetic surfaces.

One of the limitations of this study was the small sample size. However, the study’s sample size was sufficient for the objective of this study and to observe significant differences between the two groups (power = 0.88 using effect size F = 0.4, calculated by G* Power 3 [42]). Second, the participants were asked not to hold hockey sticks during their skating because of the lab’s safety issues. Not holding hockey sticks can make differences in the skaters’ skating patterns which must be further investigated. Third, comparing the kinematics of low- and high-calibre skaters on ice can support the findings of this study and must be taken as a potential future direction. Fourth, to validate the kinematics measured by the IMUs against the optical motion capture system, we had to use the sensor-to-body frame alignment based on markers. Because we had to present the measurements of both systems in the same frame and isolate only the IMUs’ orientation estimation error compared to the optical motion capture system. In the field, however, one can use a functional calibration algorithm (similar to our previous works [21,33]) that does not need recordings of an optical motion capture system. Similarly, an algorithm to detect temporal events of skating using IMUs should be used for in-field recordings. We recently proposed such algorithms in [23]. Fifth, the inconsistencies in the definition of low- and high-calibre skaters can add more complexity to inter-study comparisons. Alternatively, observational indices using the camera recordings of participants’ skating or performance questionnaires developed for on-ice skating analysis can make these comparisons more consistent.

## 5. Conclusions

The first step toward improving hockey players’ efficiency in ice hockey matches and training is accurate performance assessments, and wearable technology helps hockey coaches do this assessment in a less intrusive way. In this study, 3D joint angles of ice hockey skaters were obtained using an IMU-based wearable technology and experimentally validated their accuracy against a camera-based motion capture system. Further, a supervised learning algorithm was developed to classify low- and high-calibre skaters’ kinematics using on-ice calibre-based distinctive features. We discovered that skating on synthetic ice alters the skaters’ skating patterns such that the on-ice distinctive features could not differentiate low- and high-calibre skaters on synthetic ice with high accuracy and sensitivity. Characterizing the biomechanics of skating on synthetic ice and comparing it with on-ice skating biomechanics is important since it is an alternative to on-ice skating. The next step of this technology development would be using it on ice to analyse the player performance on ice experiments. Using the output of this technology, skating coaches and trainers can keep track of the skaters’ progress and improve their efficiency by assessing their skating performance during training sessions and matches, even remotely.

## Figures and Tables

**Figure 1 sensors-23-00334-f001:**
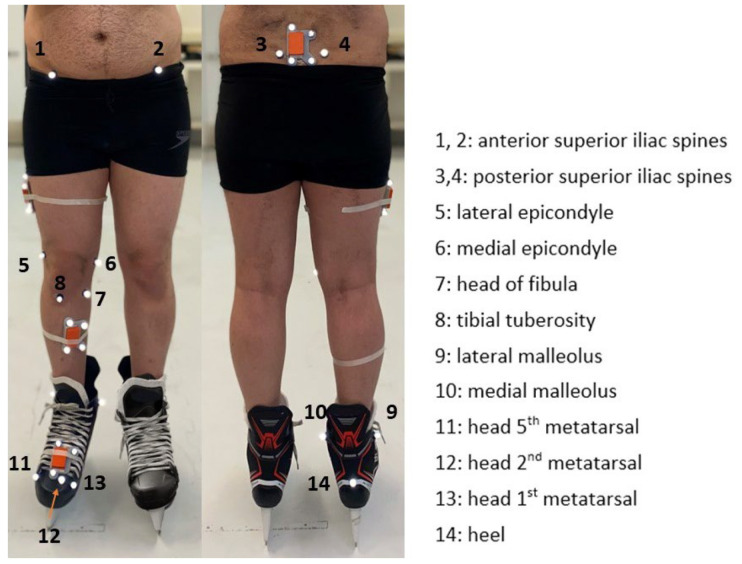
Four IMUs (orange boxes) were placed on the participants’ pelvis, thigh, shank, and skate of the dominant leg. Moreover, 14 retro-reflective were used for 3D joint angles estimation using a motion capture system.

**Figure 2 sensors-23-00334-f002:**
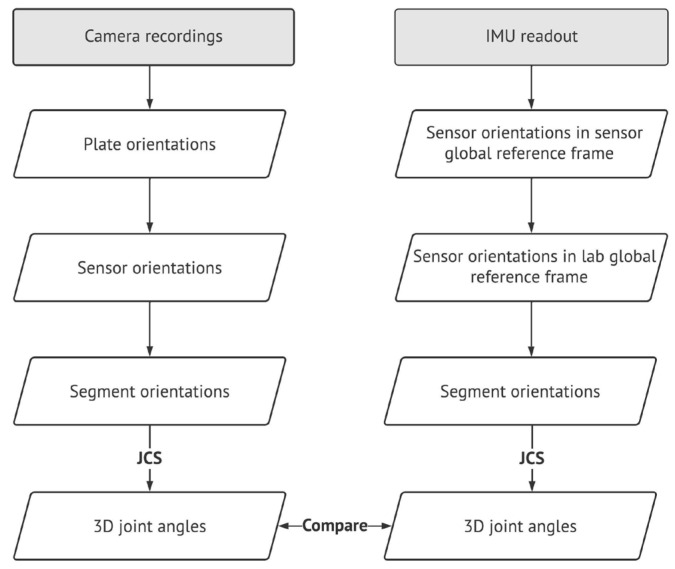
Flowchart of obtaining 3D joint angles of ice skating using IMU readouts and comparing them with the ones obtained by a stationary motion capture system.

**Figure 3 sensors-23-00334-f003:**
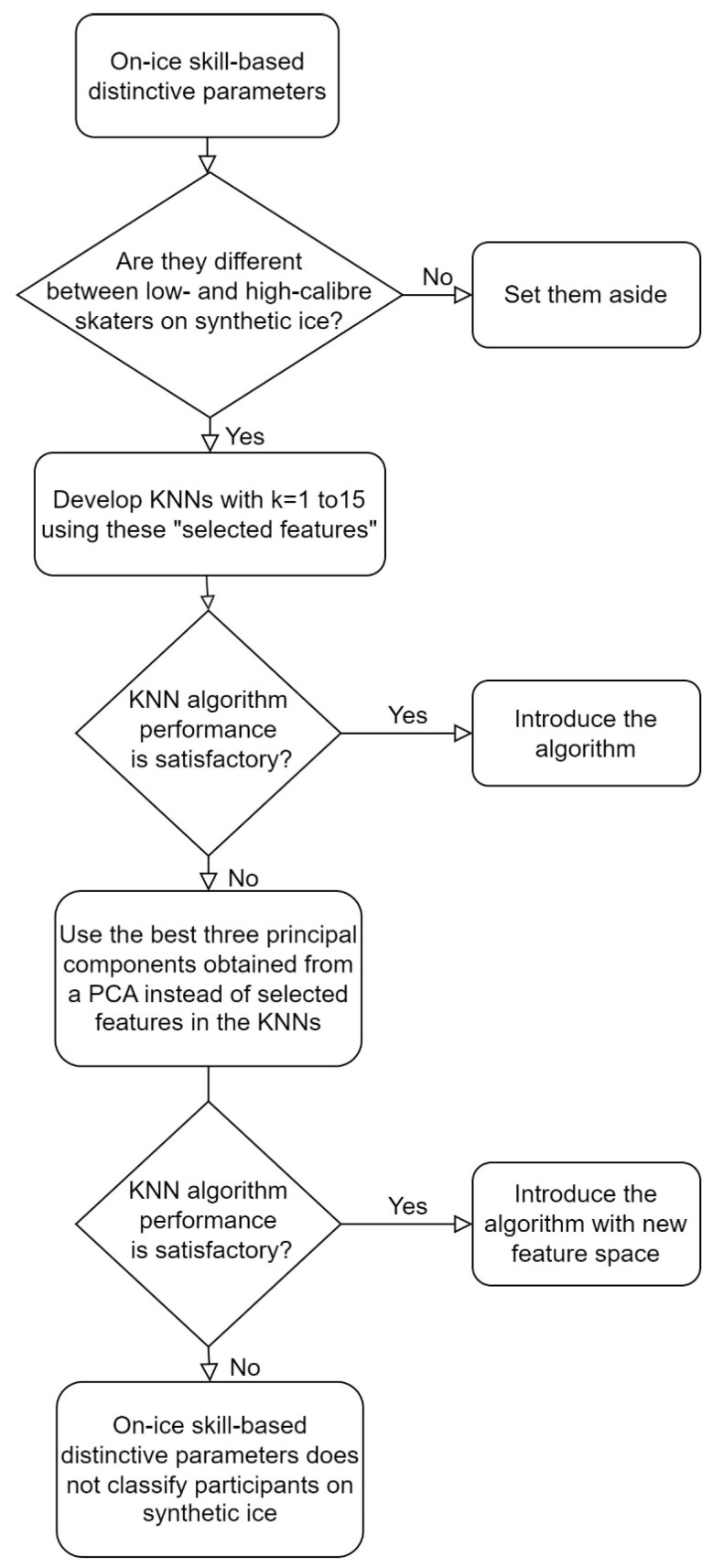
Flowchart of feature selection for KNN algorithms, checking the performance of the algorithms utilizing the selected features or principal components obtained by a PCA.

**Figure 4 sensors-23-00334-f004:**
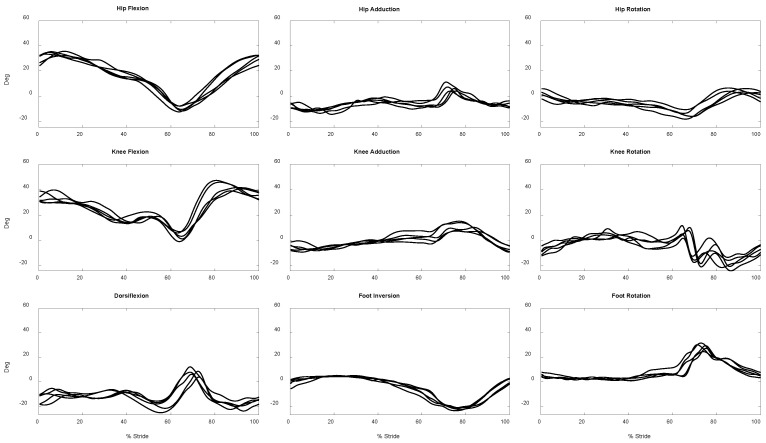
Exemplar 3D lower limb joint angles during five skating trials of a participant obtained from IMU readouts.

**Figure 5 sensors-23-00334-f005:**
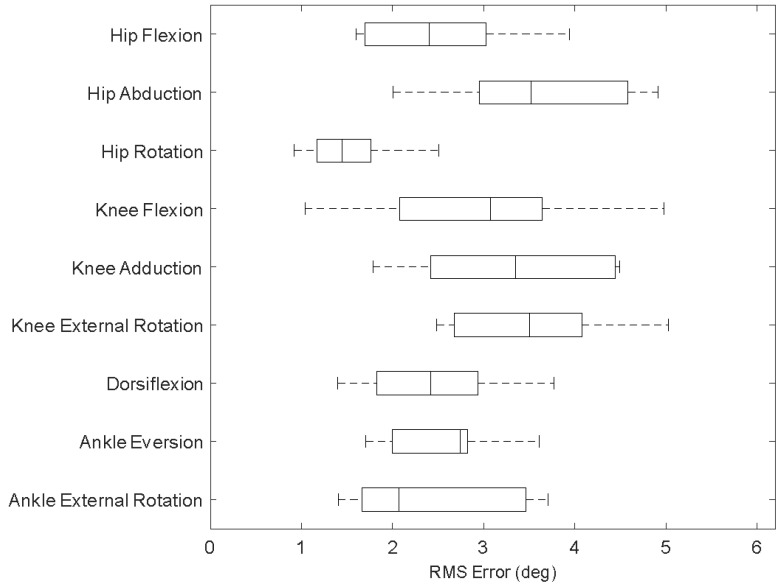
Root mean square (RMS) errors between the 3D angles obtained by IMUs readout and those obtained using camera recordings (as the reference system). First, the RMS error was attained between IMU-based and motion capture-based angles for each trial. Then, these values were averaged over all trials of each participant. Then, the RMS of all the participants’ obtained average values is presented as boxplots.

**Table 1 sensors-23-00334-t001:** On-ice distinctive features that differentiate high- and low-calibre hockey players on ice are listed here. The Friedman test was used to investigate whether they significantly differ in high- and low-calibre skating on synthetic ice experiments. The features with a *p*-value lower than or equal to 0.05 was labelled by an asterisk (*) and were used to classify high- and low-calibre skaters using KNN (Table 2).

Features	Friedman Test (*p*-Value)
Dorsiflexion range *	0.03
Ankle adduction at the end of push-off instant	0.80
Hip flexion in initial contact instant	0.67
Hip adduction in push-off instant *	0.03
Hip adduction at initial contact instant	0.15
Dorsiflexion in push-off instant	0.39
Knee flexion in push-off instant *	0.03
Hip flexion average	0.73
The interquartile range of CoM ^1^ motion ^2^ in the body mediolateral plane	0.23
Range of CoM motion in the body’s mediolateral plane	0.67
The interquartile range of CoM ^1^ motion ^2^ in the body sagittal plane	0.73
Range of CoM motion in the body sagittal plane	0.67

^1^ Centre of Mass; ^2^ CoM was obtained as the centre of the two markers placed on the posterior superior iliac spine (PSIS).

**Table 2 sensors-23-00334-t002:** The sensitivity, specificity, accuracy, and precision from a cross-validation analysis using the KNN models with k = 1 to k = 15 in classifying high- and low-calibre skaters. The feature spaces of the KNN models are either selected kinematic features of lower limb joint motions, obtained from Table 1, or the best three components obtained from the PCA on on-ice distinctive features.

	K	Sensitivity (%)	Specificity (%)	Accuracy (%)	Precision (%)
Using the selected features	1	59	86	64	75
	3	67	82	67	71
	5	60	72	58	69
	7	50	72	54	70
	9	46	70	51	70
	11	52	61	50	63
	13	50	61	47	59
	15	50	61	47	68

The three best features obtained from PCA	1	71	58	58	71
	3	61	60	55	68
	5	61	66	57	70
	7	54	68	54	69
	9	53	75	57	74
	11	48	73	53	69
	13	48	72	53	70
	15	46	78	55	74

## Data Availability

Generated datasets are available by request to the corresponding author.

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
