# Peer review of "Assessment of Three-Dimensional Kinematics of High- and Low-Calibre Hockey Skaters on Synthetic Ice Using Wearable Sensors"

_sensors, 2022, doi:10.3390/s23010334_

Round 1

Reviewer 1 Report

GENERAL COMMENTS:

This paper presents a novel idea of using wearable sensors to assess three-dimensional kinematics of high- and low-calibre hockey skaters on synthetic ice. The overall writing is good. The introduction well explained the motivation of this study, and the discussion is sufficient. The methods section needs some improvements, and there are also some specific issues that need the authors' attention.

SPECIFIC COMMENTS:

1. Line 46: it will be better to give some explanation to IMU, including the accelerometer, gyroscope, etc.

2. Line 80: How to decide high-calibre and low-calibre skaters, the authors can give more explanation.

3. Line 89: In figure 1, is it four IMU? pelvis, thigh, shank and foot?

4. Line 101: Figure 1, I recommended adding a real-world photo of the experimental setup. That will be more impressive to readers.

5. Line 108: Sensor fusion algorithms usually only estimate orientation rather than 3D joint angles.

6. Line 111: The algorithm 

7. Line 113: What is AKA?

8. Line 118: When calculating the segment orientation, does the algorithm need to use the segment frames obtained from anatomical markers? If it is, how IMU-based system can be used independently? In figure 2, can IMU readout be used for 3D joint angles without the support of optical motion capture?

9. Line 178: it will be better to add a figure of representative raw joint angles of each joint in a typical trial.

10. Line 187: How to determine the instants, like push-off instant, initial contact instant. The authors should give more explanation.

11. Figures need to be centered. Line 17-171: indent seems different.

Reviewer 2 Report

In this paper, the three-dimensional kinematics of high- and low-calibre hockey skaters on synthetic ice are evaluated using wearable sensors. This work is interesting and meaningful. However, some noticeable defects make this manuscript not convincing and scientific enough. Here, some suggestions are provided for the authors' reference.

1. Can you show the relevant information of the sensors used in the paper in more detail? Such as photos, structural dimensions, measurement principles, etc., I think this is very necessary. At the same time, I would like you to clarify whether skating with the sensor affected the experimental results.

2. Can you further clarify the definition of high- and low-calibre hockey skaters in the test?

3. There are certain differences in the body structure of male and female, which will certainly influence the experimental results to some extent. However, the gender of the test subjects is not explained in the paper, so I hope you can give a more explicit explanation.

4. The experimental flow chart in the paper is very complete, but I hope you can provide more original experimental data as support.

5. As mentioned in the paper, the sample range of 6 people in the high-level group and 6 people in the low-level group, a total of 12 people, is indeed too small. The influence of random error on the accuracy and sensitivity of the experimental results cannot be ignored. Please give a reasonable explanation that the sample range does not affect the experimental results.

6. This paper only analyzes the data of the high- and low-calibre hockey players skating on synthetic ice. In order to improve persuasiveness, I hope the author can add two experiments of skating on ice and skating on synthetic ice respectively for the high- and low-calibre hockey players, and compare them respectively.

7. Please check the syntax and formatting.
